

# Lightweight transformer image feature extraction network

Wenfeng Zheng[1], Siyu Lu[1], Youshuai Yang[1], Zhengtong Yin[2] and Lirong Yin[3]

[1] School of Automation, University of Electronic Science and Technology of China, Chengdu, Sichuan, China
[2] College of Resource and Environment Engineering, Guizhou University, Guiyang, Guizhou, China
[3] Department of Geography and Anthropology, Louisiana State University, Baton Rouge, LA, United States of America

## ABSTRACT

In recent years, the image feature extraction method based on Transformer has become a research hotspot. However, when using Transformer for image feature extraction, the model's complexity increases quadratically with the number of tokens entered. The quadratic complexity prevents vision transformer-based backbone networks from modelling high-resolution images and is computationally expensive. To address this issue, this study proposes two approaches to speed up Transformer models. Firstly, the self-attention mechanism's quadratic complexity is reduced to linear, enhancing the model's internal processing speed. Next, a parameter-less lightweight pruning method is introduced, which adaptively samples input images to filter out unimportant tokens, effectively reducing irrelevant input. Finally, these two methods are combined to create an efficient attention mechanism. Experimental results demonstrate that the combined methods can reduce the computation of the original Transformer model by 30%–50%, while the efficient attention mechanism achieves an impressive 60%–70% reduction in computation.

## INTRODUCTION

In the field of deep learning, convolutional neural networks (CNN) have been widely used for image feature extraction (*Li et al., 2022b*; *Li, Chen & Zeng, 2022*; *Liu et al., 2022*). CNNs have shown great advantages in processing images over the past decade. They can extract hierarchical representations of image data, where high-level features depend on low-level features such as edges and textures (*Tian et al., 2022*). In addition, the convolution kernel in CNN has inductive biases such as locality and translation invariance, which can capture the local details of the input image (*Zheng et al., 2017*). However, the operation of convolution inherently lacks a global understanding of the input image, and there is no way to model the dependencies between features, so that context information cannot be fully utilized. Furthermore, fixed convolutional weights lack the ability to dynamically adapt to the input changes. These limitations have motivated researchers to explore alternative models, such as Transformer, which has been successful in natural language processing (NLP) tasks. In recent years, there have been attempts to apply Transformer models to computer

Corresponding author
Lirong Yin, yin.lyra@gmail.com

vision (CV) tasks and extract features from images using Transformers (*Liang et al., 2021*; *Chen, Fan & Panda, 2021*; *Guo et al., 2021*). Because compared to CNN, Transformer can model longer-distance dependencies without being limited by local interactions. Parallel computing can also be realized, and good experimental results have been achieved in various visual tasks.

For many vision tasks, the final accuracy and time efficiency largely depend on the network responsible for extracting image features. Therefore, it is very important to design a good image feature extraction network. In the past few years, the work of designing Transformer-based image feature extraction network has begun to emerge (*Han et al., 2023*; *Khan et al., 2022*). However, any Transformer-based model has a bottleneck. That is, given a token sequence as input, the self-attention mechanism associates any pair of tokens to iteratively learn feature representation. This will lead to a quadratic relationship between the model's time and space complexity and the input tokens' amount. This quadratic complexity prevents Transformer from modelling high-resolution images, and the high computational cost poses challenges for deploying it on edge devices.

In academia, the majority of existing linear attention mechanisms aim to make the Softmax operator approximated. RFA (*Peng et al., 2021*) utilizes the stochastic Fourier characteristic theorem, and Performer (*Krzysztof et al., 2021*) utilizes positive random characteristics to approximate the Softmax operator. Nevertheless, empirical evidence supports that high sampling rates can cause instability in these methods. The way to achieve linear attention mechanisms of these methods by utilizing an effective Softmax operator approximation only within the constrained theoretical range, so if the corresponding assumptions are not fulfilled, or approximation errors accumulate, these methods may not always outperform ordinary structures.

Currently, many pruning methods for Transformer-based image feature extraction networks that operate on the token dimension involve the introduction of additional neural networks for training. To calculate the token score, and use this as a basis to further judge wither the tokens should be redundant or kept. For example, Dynamic ViT (*Rao et al., 2021*) is one of that case. However, methods in this type always reduce token by a fixed ratio at each stage. While this approach indeed externally reduces the computational burden of Transformer-based image feature extraction networks, additional computational cost is also introduced at the same time. That is, the scoring network needs to be trained in conjunction with the Transformer-based image feature extraction network and additional hyperparameters and loss items need to be added to modify the loss function. Another limitation of it is that they rely on a fixed pruning token ratio. Then when it needs to be deployed on different edge devices, the network needs to be retrained, which greatly limits the application scenarios of the model.

This study proposes a Transformer image feature extraction network based on linear attention and pruning. First, each token is individually scored externally to determine the importance of the class token used for classification based on the input tokens. Then some tokens are retained according to the score sampling, and pruning is performed from the token dimension. Then, from an internal perspective, a combination function is employed to substitute the Softmax operator in calculating the self-attention matrix, resulting in a

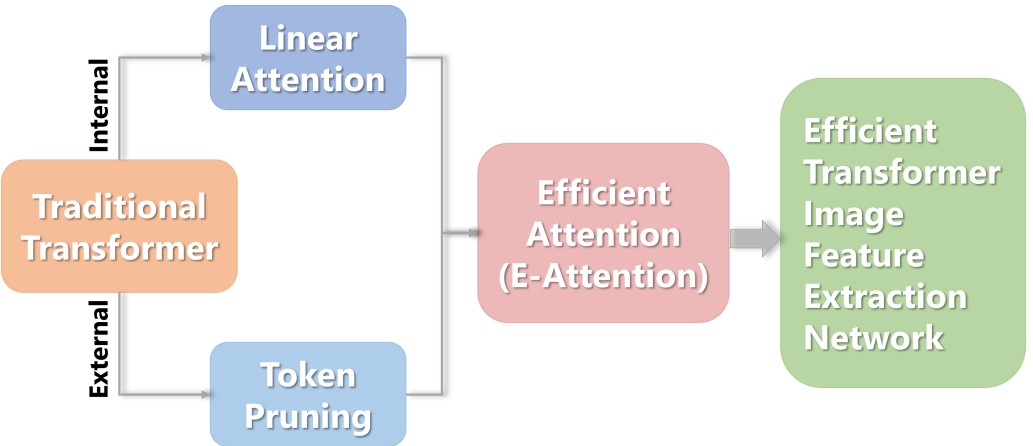

**Figure 1** The flowchart of the Transformer image feature extraction network.

new linear attention mechanism. Finally, an efficient attention mechanism (e-attention) was yielded after the two methods were merged. The flowchart is shown in Fig. 1.

## RELATED WORK

In 2017, Google proposed Transformer (*Vaswani et al., 2017*), a milestone model in the field of natural language processing. In October 2020, *Dosovitskiy et al. (2021)* proposed the ViT (VisionTransformer) model, which was the first to introduce Transformer into the visual field, proving that Transformer can also achieve good results as a network for extracting image features. From then on, the research on Transformer as an image feature extraction network has entered a rapid development process, and the development direction can be divided into two categories, namely the improvement of training strategies and models. The training strategy aspect refers to the improvement of the ViT model during the training process, while the model aspect refers to the improvement of various modules in the image feature extraction network designed based on Transformer. The current mainstream training strategy mainly refers to the model DeiT (*Touvron et al., 2021a*) proposed by Touvron in December 2020. The proposal of this model is to address the drawback of the ViT model requiring pre training using a large dataset of JFT-300M. The core of DeiT improvement is to introduce distillation learning into the training process of the Transformer model, and also provide a set of hyperparameters with good experimental results. After this, most Transformer models refer to this set of hyperparameters in the experimental process, and the hyperparameter settings in this article's experiment are also the same. At present, the improvement of self-attention mechanism can be divided into two directions: improving global attention and introducing additional local attention. (1) Improve global attention. The most typical global attention is the multi head attention in the Transformer model. When the resolution of the input image is high, the number of tokens converted is large, and the computational cost is high when calculating attention. Taking PVT (*Wang et al., 2021*) as an example, it proposes a space reduction module,

which first restores the input token sequences of each encoder to the spatial structure, then uses convolution to reduce the dimensions of the spatial structure. Finally, it is converted into token sequences and input into the encoder, reducing the matrix dimensions inside the current encoder, and then calculates attention. The design concept of MViT (*Fan et al., 2021*) proposed by Facebook in April 2021 is similar to that of PVT, and improved MViT was further proposed in December 2021 (*Li et al., 2022a*). (2) Introduce additional local attention. This refers to only calculating local windows instead of global windows when calculating attention, which can effectively reduce computational costs. After introducing local attention, it is necessary to interact with information across windows, otherwise there will be a certain degree of performance degradation. The typical algorithm for this idea is the Swin Transformer (*Liu et al., 2021*), which divides windows in advance and restricts the self attention calculation process, known as Window based Self Attention (W-MSA), which can significantly reduce the computational load. At the same time, in order to achieve interaction between different windows, the author further proposes Shifted Window Based Self Attention (SW-MSA), which shifts the window to the bottom right and alternates between different stages using W-MSA and SW-MSA. This also requires an even number of stages in the experimental process, so that local attention can be calculated within the pre-drawn window and interaction information can be cross window. Although the Swin Transformer algorithm performs well in solving the huge computational cost consumption caused by the increase in input image resolution, the internal structure design of SW-MSA is too complex and difficult to deploy. Many subsequent algorithms have proposed a large number of targeted improvements based on this, first removing SW-MSA and still requiring a global attention calculation module, which is implemented by a global attention calculation module with reduced computational complexity to achieve cross window interaction, such as Improved MViT (*Fan et al., 2021*). Another improvement is to also remove SW-MSA, but cross window information exchange is provided by specific modules proposed in the improvement paper, such as the Shuffle Transformer (*Huang et al., 2021*) proposed by Tencent in June 2021 and the MSG-Transformer (*Fang et al., 2022*) proposed by the Chinese Academy of Sciences in May 2021. From the perspective of introducing convolutional induction bias, there are also many efficient improvements, such as ViTAE (*Xu et al., 2021*), ConViT (*d'Ascoli et al., 2021*), PiT (*Heo et al., 2021*), CvT (*Wu et al., 2021*).

# METHOD

## Lightweight based on linear attention

This section internally reduces the complexity of Transformer's attention mechanism to linear, and designs a combination function to replace the original Softmax operator.

### Linear attention basics

First, mathematically describe the general form of Transformer, where the given input was represented in the embedding space as $X \in \mathbb{R}^{(n \times d)}$, denoting the transformation that the input undergoes into the Transformer module as T, thereby the definition of transforming

T is shown in Eq. (1):

$$T(\mathbf{X}) = F(Att(\mathbf{X}) + \mathbf{X}) \tag{1}$$

where F is a feedforward neural network containing residual connections; Att is a function used to compute the self-attention matrix.

The self-attention mechanism in the current Transformer model is scaling point product attention, and for the convenience of expression, the scaling factor of ATT is omitted, and the definition is shown in Eq. (2):

$$Att(\mathbf{Q}, \mathbf{K}, \mathbf{V}) = Softmax(\mathbf{Q}\mathbf{K}^{\mathbf{T}})\mathbf{V} \tag{2}$$

where $\mathbf{Q} \in \mathbb{R}^{(n \times d)}$, $\mathbf{K} \in \mathbb{R}^{(n \times d)}$, $\mathbf{V} \in \mathbb{R}^{(n \times d)}$, which can be calculated from the inputs, respectively.

An in-depth analysis of Eq. (2) shows that it is the Softmax operator defined in Eq. (2) that restricts the performance of the scaling point product attention mechanism and makes it quadratic, so we need to calculate $QK^T$ first, and a matrix of n × n was obtained, so that the complexity is O $(n^2)$ level. If there is no Softmax operator, then the three matrices $QK^TV$ will be multiplied, and using the combination law of matrix multiplication, the last two matrices $K^TV$ can be calculated first to get a matrix of $d_k \times d_v$, and then let the matrix Q to do the left multiplication, due to the actual case $d_k$ sum $d_v$ is much smaller than n, so the overall complexity can be seen as O(n) level. The Softmax operator is defined as shown in Eq. (3):

$$Softmax(z_i) = \frac{e^{z_i}}{\sum_{c=1}^{c} e^{z_c}} \tag{3}$$

where $z_i$ represents the the $i$-th node's output value, and C denotes the total number of output nodes.

The Softmax operator could convert the multiple clusters' output values to a probability distribution at range between 0 and 1, with a sum of 1. With the introduction of Eqs. (3) and (2) can be rewritten as follows, as shown in Eq. (4):

$$Att(\mathbf{Q}, \mathbf{K}, \mathbf{V})_i = \frac{\sum_{j=1}^{n} S(\mathbf{q}_i, \mathbf{k}_j) \mathbf{v}_j}{\sum_{j=1}^{n} S(\mathbf{q}_i, \mathbf{k}_j)} \tag{4}$$

To preserve the similar properties of the attention mechanism, S(·) ≥0 should be constant. This general form of attention is also known as non-local networking (*Wang et al., 2018*).

For arriving the most ideal linear level on the complexity of attention, a decomposable method of calculation is needed to effectively approximate the similarity function S(·). The associative law of matrix multiplication can be used to calculate the product of the next two moment matrices first, reducing the complexity to the linear O(n) level (*Zhuoran et al., 2021*), as shown in Fig. 2.

Most existing linear attention mechanisms aim to approximate the Softmax operator estimation, but these methods are sensitive to the choice of sampling rate. Hence, it is worth considering whether a decomposable similarity function can be directly employed

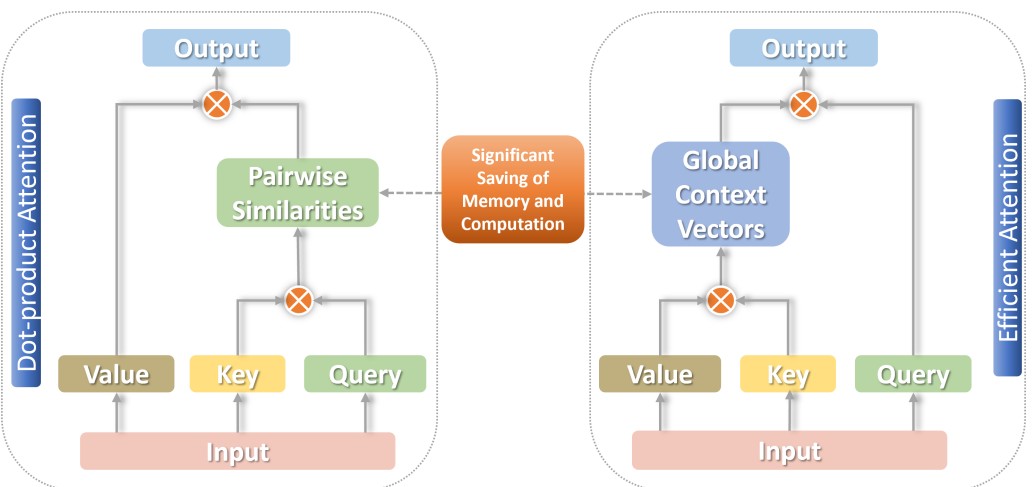

**Figure 2** Traditional self-attention *versus* linear self-attention.

as a replacement for the Softmax operator under the premise of ensuring the experimental effect. To achieve this, it is necessary to identify the crucial characteristics of the current attention mechanism and design a decomposable similarity function that fulfills the prerequisites for linear attention.

### Alternative function design

Through an in-depth reading of literature research, two critical properties that significantly impact the performance of current attention mechanisms are summarized. They are non-negative elements in the attention matrix (*Katharopoulos et al., 2020*; *Tsai et al., 2019*) and non-linear reweighting schemes (*Zhen et al., 2022*; *Paoletti et al., 2023*; *Baldi & Vershynin, 2023*).

First, only positive values are kept in the attention matrix, and features with negative correlation are ignored, thus effectively avoiding aggregation of irrelevant contextual information. Second, the nonlinear reweighting mechanism can make a focus on the distribution weights of attention, thereby stabilizing the training process. It can also help the model to amplify the locality, which means that a large part of the context dependence comes from neighboring markers.

Combining the above conclusions, this subsection proposes a combination function to replace Softmax. This combination function satisfies the above two properties and consists of two sub-functions. are used to implement non-negativity and non-linear reweighting, respectively, as shown in Eq. (5):

$$\text{Att}(\mathbf{Q}, \mathbf{K}, \mathbf{V}) = g(f(\mathbf{Q}, \mathbf{K}))\mathbf{V} \tag{5}$$

In the formula, $f(\cdot)$ and $g(\cdot)$ are two self-defined functions to ensure the above two properties, defined as shown in Eqs. (6) and (7):

$$f(x) = \begin{cases} x+1 & x \geq 0 \\ e^x & x < 0 \end{cases} \tag{6}$$

$$g(\mathbf{q}_i, \mathbf{k}_j) = \mathbf{q}_i \mathbf{k}_j^T \sin\left(\frac{\pi(i+n-j)}{2n}\right) \tag{7}$$

where $i, j = 1, \ldots, n$.

Equation(6) is used to ensure the non-negativity of the attention matrix.

Equation (7) is formulated in this form for several reasons:

(1) Realize local deviation. When i and j are close, the corresponding trigonometric function value is close to 1. When i and j are far away, the difference between the two is close to n, and the corresponding trigonometric function value is close to 0, so that the corresponding The similarity function is negligible;

(2) The trigonometric function itself is nonlinear, so the design can focus on the distribution of weights, so as to achieve the purpose of stabilizing the training process;

(3) Eq. (7) can be decomposed by the sum-difference product formula, so that the associative law of matrix multiplication can be used to reduce the complexity of the attention mechanism to linear.

The Eq. (8) is the dismantling process of the sub-function $g(\cdot)$ for satisfying the non-linear reweighting property:

$$
\begin{aligned}
g\left(\mathbf{q}'_i, \mathbf{k}'_j\right) &= \mathbf{q}'_i \mathbf{k}'_j T \sin\left(\frac{\pi(i+n-j)}{2n}\right) \\
&= \mathbf{q}'_i \mathbf{k}'_j T \sin\left(\frac{\pi(i-j)}{2n} + \frac{\pi}{2}\right) \\
&= \mathbf{q}'_i \mathbf{k}'_j T \left(\cos\left(\frac{\pi i}{2n}\right)\cos\left(\frac{\pi j}{2n}\right) + \sin\left(\frac{\pi i}{2n}\right)\sin\left(\frac{\pi j}{2n}\right)\right) \\
&= \left(\mathbf{q}'_i \cos\left(\frac{\pi i}{2n}\right)\right)\left(\mathbf{k}'_j T \cos\left(\frac{\pi j}{2n}\right)\right) + \left(\mathbf{q}'_i \sin\left(\frac{\pi i}{2n}\right)\right)\left(\mathbf{k}'_j T \sin\left(\frac{\pi j}{2n}\right)\right)
\end{aligned} \tag{8}
$$

Among them, $\mathbf{q}'_i = f(\mathbf{q}_i)$, $\mathbf{k}'_j = f(\mathbf{k}_j)$, $f(\cdot)$ are shown in Eq. (2).

Let $\mathbf{q}_i^{cos} = \mathbf{q}'_i \cos\left(\frac{\pi i}{2n}\right)$, $\mathbf{q}_i^{sin} = \mathbf{q}'_i \sin\left(\frac{\pi i}{2n}\right)$, $\mathbf{k}_j^{cos} = \mathbf{k}'_j T \cos\left(\frac{\pi j}{2n}\right)$ and $\mathbf{k}_j^{sin} = \mathbf{k}'_j T \sin\left(\frac{\pi j}{2n}\right)$.

Apparently, $\mathbf{q}_i^{sin}$ and $\mathbf{q}_i^{cos}$ are transformed from the $i$-th column vector $\mathbf{q}_i$ of $Q$ through self-defined non-negative function $f(\cdot)$ and sine or cosine function. Denote the $Q$ after this transformation as $Q^{cos}$ and $Q^{sin}$. Similarly, $K$ after transformation is $K^{cos}$ and $K^{sin}$. Thus, the final expression after linear decomposition is obtained, as shown in Eq. (9):

$$\text{Att}(\mathbf{Q}, \mathbf{K}, \mathbf{V}) = g(f(\mathbf{Q}, \mathbf{K}))\mathbf{V} = \mathbf{Q}^{cos}\left(\mathbf{K}^{cos}\mathbf{V}\right) + \mathbf{Q}^{sin}\left(\mathbf{K}^{sin}\mathbf{V}\right) \tag{9}$$

The attention calculation Eq. (9) proposed in this section can make use of the associative law of matrix multiplication to minimize the computational complexity of the linear attention mechanism. Figure 3 shows the calculation process.

## Lightweight based on token pruning

This section prunes the Transformer-based image feature extraction network from the token dimension to reduce the computational cost. Tokens converted from different input

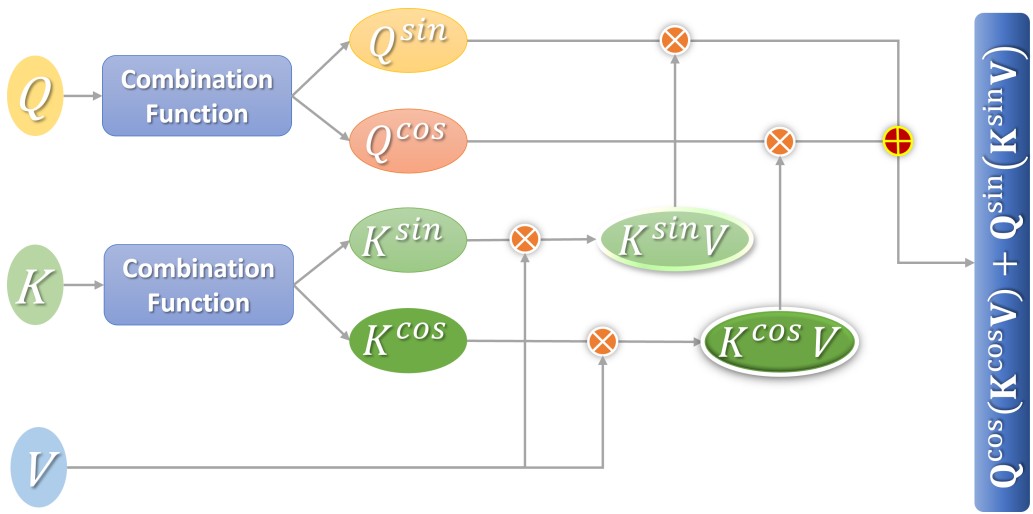

**Figure 3 Flow chart of linear attention calculation.**

images are first estimated with their respective scores, and then part of the tokens are retained through sampling.

### *Score*

After bypassing the path of training additional parameters to obtain the score of the input token, we can focus on the existing information of the image feature extraction network itself designed based on Transformer. All tokens input by the self-attention layer inside the model are divided into two parts. The first part only includes the first class token, which is additionally introduced to judge the category of the input image in the final stage, and it is placed in the first position of the token sequence. The second part is all the remaining tokens, which are obtained by dividing and transforming the input image.

To prune from the token dimension is to reduce the second part of tokens converted from images as much as possible and record the number of tokens in this part as $m$. That is to say, there are $m+1$ tokens in the input, and there are $r'+1$ tokens after the output. Here $r'$ is the number of tokens in the second part retained after pruning, and the corresponding data range is $r' \leq R \leq m$. Here $R$ is a parameter given in advance, and its function is to control the maximum number of tokens retained after sampling.

In the standard self-attention layer, $\mathbf{Q} \in \mathbb{R}^{((m+1)\times d_q)}$, $\mathbf{K} \in \mathbb{R}^{((m+1)\times d_k)}$, $\mathbf{V} \in \mathbb{R}^{((m+1)\times d_v)}$ is calculated from the input token, which is recorded as $\mathbf{X} \in \mathbb{R}^{((m+1)\times d)}$. The self-attention matrix $A$ is calculated from both $Q$ and $K^T$. It should be noted that the attention calculation method at this time is still the standard scaling dot product self-attention. The attention matrix captures the association's similarity or degree among all input tokens.

For example, in the self-attention matrix A, the elements of [i, j] or [j, i] represent the correlation degree between the $i$-th and the $j$-th tokens among all the currently input tokens. The larger the value, the more related the two are, indicating that each other is more important to another token.

It can be inspired from this, since the first class token of the currently input token sequence is used for the classification of the final stage. Then in the self-attention matrix, if other tokens are more correlated with the first class token used for classification, it means that these tokens are more important for classifying the class token of the input image category at least at the current stage. To measure the similarity between these tokens and the first-class token, all values except the 1st one in the 1st row of the self-attention matrix are utilized. Therefore, when designing the score formula for calculating the importance of each token except class token, the calculation method used is shown in Eq. (10):

$$h_j = \frac{a_{1,j}}{\sum_{i=2}^{m} a_{1,i}} \tag{10}$$

In the Equation, $a_{1,j}$ and $a_{1,i}$ represent the elements of the 1st row, column $j$ and column $i$ respectively of the self-attention matrix A, $h_j$ is the $j$-th token's score.

In addition, for the multi-head attention layer, each head's score can be calculated separately, and then all the heads can be added together.

### Sampling

In the previous section, the importance scores of each token have been obtained in a parameterless way. Now some tokens can be removed from the self-attention matrix based on these scores. It is natural to think of first screening out some tokens with the highest scores, keeping them, and then directly removing the remaining tokens with lower scores.

But this method has two problems: First, the number of tokens that are screened out is not easy to determine, and it is difficult to achieve adaptive dynamic changes based on different input images. Secondly, if tokens with lower scores are directly screened out at a certain stage from the beginning, then these screened out tokens are not necessarily unimportant for the final classification. Just as the CNN-based image feature extraction network extracts only shallow features such as edges and textures at the beginning, but it extracts more advanced semantic features later. Different tokens may have different functions at different stages and represent different meanings. The current score is only obtained at the current stage. If it is simply deleted based on the low effect of an intermediate stage, the deleted token will not be able to enter the subsequent stage. But it may play an important role in a later stage, which will affect the final experimental results instead.

Therefore, according to these scores, some tokens can be randomly selected to be retained by sampling. If the score is lower, it has a lower probability of being selected at the current stage and thus cannot be retained, and vice versa. This method is indeed better than crudely deleting tokens with lower scores in terms of experimental results. This randomness can offset the deficiency of directly deleting token to a certain extent.

Then it needs to be sampled according to the scores of these tokens. Tokens with higher scores have a greater probability of being retained, and tokens with lower scores have a lower probability of being retained. That is to find a probability distribution that obeys these scores, and then sample according to this probability distribution. The cumulative distribution function corresponding to these scores can be calculated and inverse transformed to obtain the inverse function of this random distribution.

First, cumulative distribution function (CDF) can characterize the probability distribution of a random variable x, and it is denoted as X. For all real numbers x, the cumulative distribution function is defined (*Han et al., 2021*) as shown in Eq. (11):

$$F_X(x) = P(X \le x). \tag{11}$$

If the cumulative distribution function $F$ is a continuous strictly increasing function, then there is a corresponding inverse function $F^{-1}(y)$, and the range of $y$ is [0,1). The inverse function can be utilized to produce random variables that follow this random distribution. That is, if take $F_X(x)$ as the CDF of $X$, and there is an inverse function $F_X^{-1}$. If $a$ is a random variable with the range of $[0,1)$ and has a uniformly distribution, then $F_X^{-1}(a)$ obeys the $X$ distribution. Therefore, the CDF can be calculated according to the following formula, as shown in Eq. (12):

$$CDF_i = \sum_{j=2}^{i} h_j. \tag{12}$$

Note here that the accumulation starts from the second token, because the first classification class token must be retained. After obtaining the CDF, the sampling function can be obtained according to its inverse form, as shown in Eq. (13):

$$\eta(v) = CDF^{-1}(v) \tag{13}$$

where $v$ is between 0 and 1.

Specifically, the sampling strategy in this section is to randomly select a number $v$ from the uniform distribution between 0 and 1. Then calculate the corresponding $\eta(v)$, and then select the nearest integer for sampling. This operation needs to be performed a fixed number of $R$ times. In these $R$ samplings, a token may be selected multiple times, so that the number of tokens actually sampled $r'$ is less than or equal to the fixed $R$.

After sampling the tokens to be retained, the initial attention matrix A changes from the initial $m+1$ row to the current $r'+1$ row, and then adds it to the next calculation process.

## Fusion of linear attention and token pruning

The core of the linear attention mechanism is to use the associative law of matrix multiplication to first calculate the product of the last two matrices $K^T V$ among the three matrices $QK^T V$, and then multiply the matrix $Q$ to the left. In the token pruning module, the token score converted from the input image uses the attention matrix after multiplying the two matrices of $QK^T$ and undergoing nonlinear transformation. The two are inherently in conflict and seem incompatible.

However, since the score uses the association degree among the 1st class token and other tokens, that is, the first row of the self-attention matrix excludes all values except the first one. Inspired by this, when the linear attention mechanism calculates the product of the last two matrices $K^T V$ among the three matrices $QK^T V$, the first column elements of the matrix $K^T$ are reserved first. Then multiply each row of the matrix $Q$ (except the 1st row) with the 1st column of $K^T$ and add them together. In this way, while ensuring the linear attention mechanism, it is also possible to additionally calculate the association degree

between the class token and other tokens. This process is linear, so it will not increase the amount of calculation too much.

After sampling the tokens to be retained, the token pruning method is to change the attention matrix A calculated by $QK^T$ from the initial $m+1$ rows to $r'+1$ rows. Here you can change the matrix Q from the original $m+1$ row to the current $r'+1$ row, and then add it to the next calculation process. The attention calculation method after the fusion of the two methods is shown in Eq. (14):

$$\text{Att}\left(\mathbf{Q}',\mathbf{K},\mathbf{V}\right) = g\left(f\left(\mathbf{Q}',\mathbf{K}\right)\right)\mathbf{V} = \mathbf{Q}'^{\cos}\left(\mathbf{K}^{\cos}\mathbf{V}\right) + \mathbf{Q}'^{\sin}\left(\mathbf{K}^{\sin}\mathbf{V}\right). \tag{14}$$

A more efficient attention mechanism (e-attention) is obtained after the fusion. Figure 4 shows the calculation process.

## Dataset

This study is based on the comparative experiment verification of the ImageNet1k dataset, which is widely used in the field of image classification, and the COCO dataset, which is commonly used in the field of target detection.

The ImageNet dataset is currently a dataset widely used in the field of artificial intelligence images (*Yang et al., 2022*). Most of the work on image positioning, classification, and detection is based on imageNet. It is widely used in computer vision, maintained by the team at Stanford University, and easy to use. ImageNet contains more than 14 million images with more than 20,000 classification categories. The ISLVRC competition uses a lightweight version of the ImageNet dataset. In some papers, this light version of the data is called ImageNet 1K, which means that there are 1,000 categories.

The COCO dataset is a large-scale detection and segmentation dataset maintained by Microsoft, which is mainly obtained in complex daily scenes (*Lin et al., 2014*). The COCO dataset mainly solves the three problems of contextual relationship between targets, target detection and two-dimensional precise positioning. It contains 91 categories. Although the number of categories is much smaller than ImageNet, each category contains a very large number of pictures, and more specific scenes in each category can be obtained. Containing 200,000 images with more than 500,000 annotations for 80 of the 91 categories, it is arguably the most extensive public object detection dataset. COCO contains 20G pictures and 500M labels, and the ratio of the datasets (training, test, and validation) is 2:1:1.

# EXPERIMENT

## Environment and hyperparameter settings

The environment of the experiment is shown in Table 1.

The hyperparameter settings of all experimental models adopt the hyperparameters provided by the model DeiT (*Touvron et al., 2021a*). It can achieve up-points without changing the structure of the ViT model. As shown in Table 2.

## Evaluation indicators

The methods proposed in this study are all aimed at accelerating the model, and the complexity and speed of the model need to be considered. Therefore, the three indicators

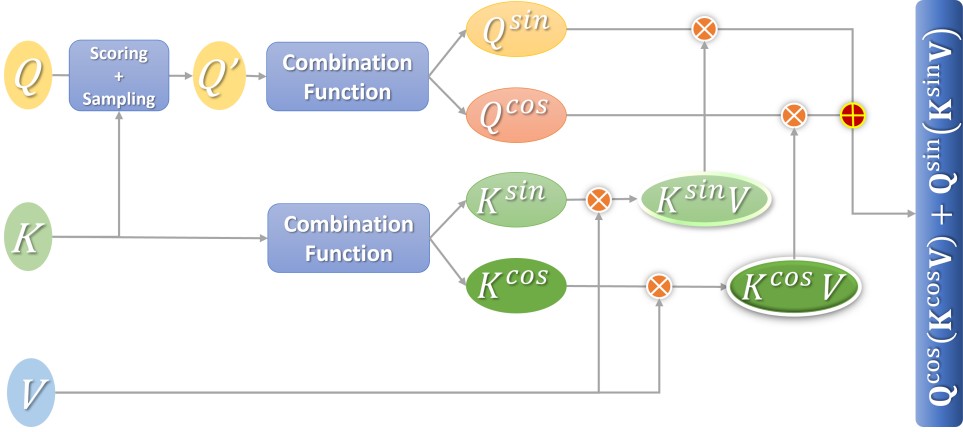

**Figure 4**  Flowchart of efficient attention calculation.

**Table 1**  The environment of the experiment.

| Name | Version |
| --- | --- |
| Editor | VS-code |
| System environment | Windows 10 education edition 64-bit system with 32 GB of memory |
| Processor | Intel(R) Core (TM) i79700K |
| Graphics processor | Nvidia GeForce RTX 2080Ti 11 GB |
| Software | Python3.6 |
| Neural network library | Pytorch |

of Params, FLOPs and FPS are selected. On the other hand, to evaluate the model's performance, the two indicators of Top-1 Acc and mAP were also selected.

In the image classification task of computer vision, evaluating the quality of a classification model is mainly judged by the accuracy and error rate identified by the classification model. The error rate includes Top-5 error and Top-1 error, and the accuracy rate includes Top-5 and Top-1. This study selects the Top-1 accuracy indicator to test the accuracy of model classification after introducing the two methods of linear attention mechanism and token pruning. On the other hand, this study selects the FLOPs index to describe the calculation amount of the model.

In the target detection task, the quality of a detection model is mainly evaluated through the mAP value. On the other hand, when using various methods to speed up the detection model and reduce its complexity, the corresponding performance indicators generally use FPS and Params.

## Benchmark model

In the experiment, six selected benchmark models including DeiT (*Touvron et al., 2021a*), PVT (*Wang et al., 2021*), Swin Transformer (*Huang et al., 2021*), TNT (*Han et al., 2021*), T2T-ViT (*Yuan et al., 2021*) and CaiT (*Touvron et al., 2021b*). In the specific experiment, the different sizes of the six models were also compared with each other, to further analyze

**Table 2    The hyperparameter settings of all experimental models.**

| Hyperparameter | Value/Choice |
| --- | --- |
| Epochs | 300 |
| Batch size | 1024 |
| Base learning rate | 0.0005 |
| Optimizer | AdamW |
| Learning rate decay strategy | Cosine |
| Weight decay | 0.05 |
| Dropout | 0.1 |
| Warmup epochs | 5 |
| Pruning method K | 70% |

the effect of these three innovations. A brief explanation will be provided for each of these six models.

(1) DeiT: The model DeiT was proposed to alleviate the limitations of traditional Transformer models, which require good performance and generalization only on a large dataset JFT-300 containing three million images. The author of DeiT proposed a new training scheme combining the lightweight operation of distillation, which introduces distillation tokens to enable students to learn from the teacher through attention, and proposes a Transformer specific teacher-student strategy. When the DeiT model is only trained on ImageNet, a convolution free Transformer with good performance is obtained. During the training of DeiT, the prediction vectors of the additional class token and distillation token introduced for classification are averaged before being converted into probability distributions.

(2) PVT: The model PVT was proposed to overcome the difficulties encountered by traditional visual Transformer models when applied to different dense prediction tasks. It is the first visual Transformer model that can handle dense prediction tasks with different resolutions. Compared to ViT, which used to consume a lot of computation in the past, it can use progressive pyramid reduction to reduce computational costs.

(3) Swin Transformer: The model Swin Transformer proposes a layered Transformer that utilizes moving windows to calculate feature representations, in order to address the significant differences in visual entity resolution. During the internal design of the model, non overlapping local windows are first divided in advance, limiting the calculation of self attention to them, and combining cross window connections to improve model efficiency. This design can model images of different resolutions and significantly reduce computational costs.

(4) TNT: The model TNT was proposed because traditional Transformer models ignore the inherent information of the patches converted from input images. Therefore, TNT models both patch level and pixel level feature representations.

(5) T2T ViT: The model T2T ViT was proposed because ViT cannot model important local structures such as edges and lines between adjacent pixels. T2T-ViT models the local structure around tokens by recursively aggregating adjacent objects layer by layer, resulting

**Table 3  Comparison of linear attention mechanisms.**

| Different linear attention | Complexity | FLOPs | Top-1 ACC |
|---|---|---|---|
| DeiT-S | $O(n^2)$ | 4.6 | 79.8 |
| Linformer | $O(n)$ | 2.4 | 78.7 |
| Performer | $O(n)$ | 2.6 | 76.8 |
| Nyströmformer | $O(n)$ | 2.4 | 79.3 |
| Ours-A | $O(n)$ | 2.1 | 77.1 |
| Ours-B | $O(n)$ | 2.3 | 76.2 |
| Ours | $O(n)$ | 2.3 | 78.8 |

in multiple adjacent tokens being aggregated into a single token. At the same time, it reduces the length of token sequences and computational costs.

(6) CaiT: CaiT makes it easier for deeper Transformers to converge and improve accuracy. A new and efficient method for processing classified tokens has been proposed. Two changes were made to the Transformer architecture, significantly improving the accuracy of deep transformers.

## Experimental exploration of linear attention

In this subsection, we first use different linear attention models to conduct image classification experiments on the ImageNet1k dataset. In this way, the benefit and drawback of each linear attention mechanism are compared, and the results are presented in Table 3.

The improvement of this work is derived from two key properties that influence Softmax: non-negative and non-linear reweighting of attention matrix elements.

We first consider only keeping the non-negativity of the attention matrix, and abandoning the non-linear reweighting scheme. Therefore, change (Eq. 15) to (Eq. 16), so that the new attention mechanism that only retains the non-negativity of the attention matrix is denoted as Ours-A.

$$g\left(\mathbf{q}_i, \mathbf{k}_j\right) = \mathbf{q}_i \mathbf{k}_j^{\mathrm{T}} \sin\left(\frac{\pi(i+n-j)}{2n}\right) \tag{15}$$

$$g\left(\mathbf{q}_i, \mathbf{k}_j\right) = \mathbf{q}_i \mathbf{k}_j^{\mathrm{T}}. \tag{16}$$

The second consideration is to keep only the non-linear reweighting scheme and discard the non-negativity of the attention matrix. Therefore, Eq. (17) is changed to Eq. (18), resulting in a new attention mechanism Ours-B that only retains the property of the non-linear reweighting scheme.

$$f(x) = \begin{cases} x+1 & x \geq 0 \\ e^x & x \vert 0 \end{cases} \tag{17}$$

$$f(x) = x. \tag{18}$$

Finally, the two new attention mechanisms that retain only one feature are compared with the attention mechanism that retains two features proposed in this study. Use these three attention mechanisms to replace the original Transformer model DeiT-S, and the resulting three new Transformer models. Using these three models and the Transformer model without any changes at the beginning to do image classification experiments on the ImageNet1k dataset, as shown in Table 3.

It can be seen from Table 3 that compared with DeiT-S, the linear Transformer could greatly decrease the calculation amount FLOPs of forward reasoning, and all of them have increased by about 50%. And the linear attention mechanism proposed in this study has improved the most on the FLOPs indicator. In terms of accuracy, the classification accuracy is second only to Nyströmformer. Based on the two indicators, the proposed linear attention mechanism has a greater advantage compared with the same type of linear attention mechanism.

The linear attention mechanism obtained by retaining two characteristics has an obvious higher accuracy than these retaining only one characteristic. Meanwhile, the FLOPs index is still greatly improved when only one characteristic is retained, as the attention mechanism changed to linear.

## Token pruning internal experiment research

(1) Comparison of different scoring methods

To evaluate the effect of this method, two other methods were selected to compare. The all score is adding all tokens' attention weights to find some of the most important tokens. The random token is to select another token randomly (not classification token), the score was calculated according to its attention weight. The obtained experimental results are shown in Fig. 5.

As shown in Fig. 5, the class token's attention weight is better, indicating that it is more valuable for evaluating candidate tokens. The reason is that the class token will be utilized for predicting class probabilities during the final stage. Therefore, the attention weights corresponding to class tokens indicate which tokens have a greater impact on the final output of tokens for classification. Summing up all attention weights merely reveals the token with the highest attention weight among all others, which may not necessarily be valuable for token classification purposes. Finally, a token is randomly selected, and the final score is calculated with reference to its attention weight, which is the worst.

(2) Comparison of different sampling methods

This section is based on the inverse function to achieve downsampling of input tokens with different scores. The comparison is shown in Fig. 6.

Obviously, the method adopted in this section is better than directly selecting the first k tokens.

## RESULTS

### Experimental results of image classification

The experiment in this section is to take the average of three repeated experiments on the ImageNet1k dataset for the different benchmark models for verification. To ensure that

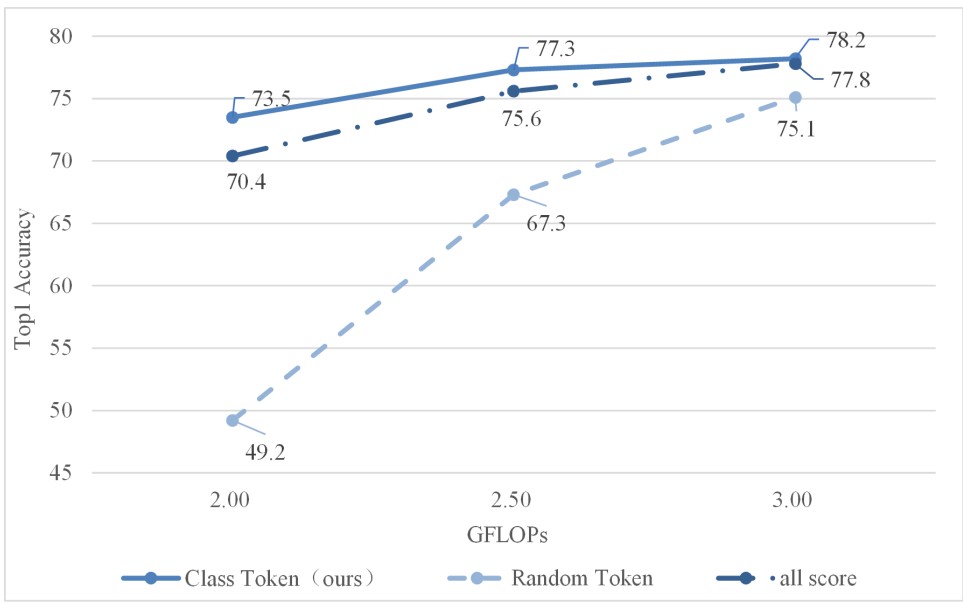

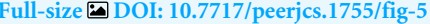

**Figure 5  Comparison of different scoring methods.**

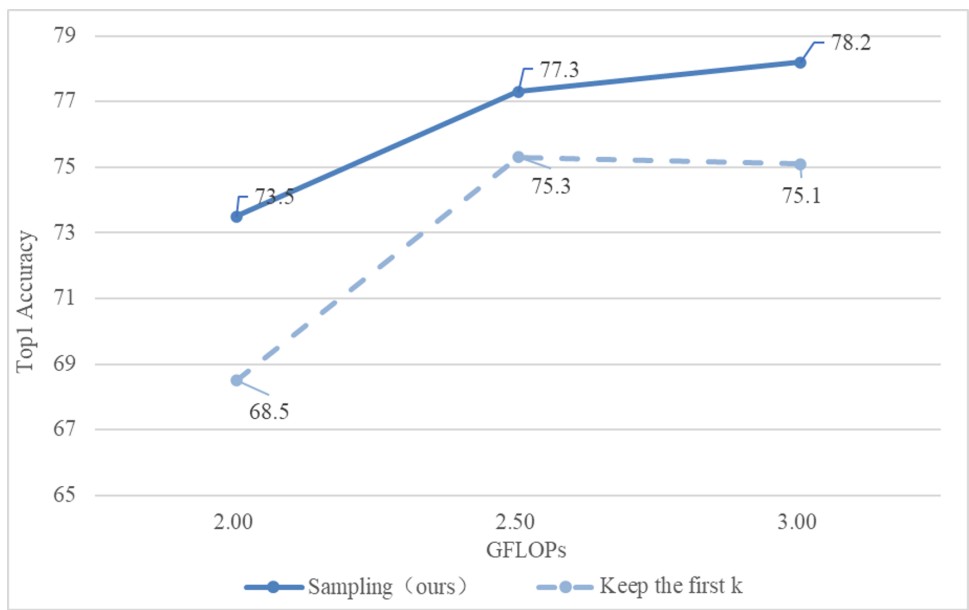

**Figure 6  Comparison of different sampling methods.**

the experimental results are as accurate as possible, the errors caused by uncertain factors are minimized. The experimental results are shown in Table 4.

Next, select the six models in the table above. Add these six different Transformer models to token pruning, linear attention, or e-attention, and then compare the FLOPs indicators. Table 5 shows the results.

**Table 4   Experimental results of benchmark model image classification.**

| Model | Top-1 accuracy | Params (M) | FLOPs(G) |
|---|---|---|---|
| DeiT-S | 79.8 | 22 | 4.6 |
| PVT-S | 79.8 | 25 | 3.8 |
| Swin-T | 81.3 | 29 | 4.5 |
| TNT-S | 81.5 | 24 | 5.2 |
| T2T-ViT-14 | 81.5 | 22 | 5.2 |
| CaiT-XXS36 | 79.7 | 17 | 3.8 |
| PVT-M | 81.2 | 44 | 6.7 |
| Swin-S | 83.0 | 50 | 8.7 |
| T2T-ViT-19 | 81.9 | 39 | 8.9 |
| CaiT-XS36 | 82.9 | 38 | 8.1 |
| DeiT-B | 81.8 | 86 | 17.5 |
| PVT-L | 81.7 | 62 | 9.8 |
| Swin-B | 83.3 | 88 | 15.4 |
| TNT-B | 82.9 | 66 | 14.1 |
| T2T-ViT-24 | 82.3 | 64 | 14.0 |
| CaiT-S36 | 83.9 | 68 | 13.9 |

**Table 5   Comparison of FLOPs indicators.**

| Model | DeiT-S | PVT-L | Swin-B | TNT-B | T2T-ViT-24 | CaiT-S36 |
|---|---|---|---|---|---|---|
| Base | 79.8 | 81.7 | 83.3 | 82.9 | 82.3 | 83.9 |
| Linear attention | 78.8 | 80.5 | 81.8 | 81.3 | 80.2 | 82.2 |
| Token pruning | 79.6 | 81.5 | 83.1 | 82.7 | 82.0 | 83.7 |
| E-Attention | 78.6 | 80.4 | 81.4 | 81.1 | 80.0 | 82.1 |

For the six selected models, no matter which improvement method is introduced alone, or the two methods are introduced together, the FLOPs index has a large improvement. Because these two methods were originally designed to accelerate the model from both the external and internal perspectives of the model. For the model DeiT-B, after changing the attention mechanism to linear alone, the FLOPs indicator increased by 52%. After introducing the pruning module alone, the FLOPs index increased by 32%. When both are incorporated into the model improvements, the total improvement is 63%.

Specifically, for the remaining five models of PVT-M, Swin-S, TNT-S, T2T-ViT-19 and CaiT-XS36, the improvement after changing the attention mechanism to the linear attention mechanism alone is 49%, 51%, 48%, 53%, and 52%. The improvement after introducing the model lightweight token pruning module alone is 33%, 35%, 33%, 31%, and 32%. After both improvements are added, the improvement of the model is 62%, 62%, 59%, 65%, and 62%.

Next, select the six models in the table above. Add these six different Transformer models to token pruning, linear attention or e-attention, and then compare the Top-1 Accuracy indicators. As Table 7 shows below.

**Table 6  Comparison of Acc indicators.**

| Model | DeiT-B | PVT-M | Swin-S | TNT-S | T2T-ViT-19 | CaiT-XS36 |
|---|---|---|---|---|---|---|
| base | 17.5 | 6.7 | 8.7 | 5.2 | 8.9 | 8.1 |
| Linear attention | 8.4 | 3.4 | 4.3 | 2.7 | 4.2 | 3.9 |
| Token pruning | 11.9 | 4.4 | 5.7 | 3.5 | 6.1 | 5.5 |
| E-Attention | 6.5 | 2.6 | 3.3 | 2.1 | 3.1 | 3.1 |

**Table 7  Experimental results of target detection.**

| Model | mAP | Params (M) | FPS |
|---|---|---|---|
| DeiT-S | 43.6 | 35 | 8.5 |
| DeiT-S+ linear | 42.9 | 35 | 12.1 |
| DeiT-S+ pruning | 43.3 | 35 | 11.1 |
| DeiT-S+ E-Attention | 42.3 | 35 | 13.6 |
| Swin-T | 47.0 | 39 | 6.3 |
| Swin-T+ linear | 46.3 | 39 | 9.5 |
| Swin-T+ pruning | 46.7 | 39 | 8.2 |
| Swin-T+ E-Attention | 45.7 | 39 | 10.1 |

No matter which improvement method is introduced alone, or the two methods are introduced together, the Top-1 Accuracy indicator has a certain degree of decline. Because compared with the original model, the two methods of model improvement are to speed up the model, and both are to increase the speed of the model at the expense of part of the performance.

Specifically, for the six models, the decline after changing the attention mechanism to a linear attention mechanism is 1.3%, 1.5%, 1.8%, 1.9%, 2.5%, and 2.0%. After the model lightweight token pruning module is introduced separately, the declines are 0.2%, 0.2%, 0.3%, 0.2%, 0.4%, and 0.3%. After both improvements are added, the decline of the model is 1.5%, 1.6%, 2.3%, 2.2%, 2.8%, and 2.2%.

## Experimental results of target detection

The experiments in this section use the Transformer-based target detector Deformalbe DETR to train on the COCO dataset. Then replace its feature extraction part with models DeiT-S and Swin-T, without any modification elsewhere. Then add token pruning, linear attention or e-attention on the basis to compare the indicators mAP and FPS. As shown in Table 6:

No matter whether the attention mechanism is improved to be linear or token pruning is introduced, or the two are included together, the change rule of the index FPS is the same as the index FLOPs of the image classification experiment. After changing the attention mechanism to linear alone, the FPS indicator is increased by about 50%–60%. After introducing the pruning module alone, the FPS index is increased by about 30%–40%. After introducing both improvements at the same time, the improvement ranges from 60% to 70%.

# DISCUSSION AND CONCLUSION

Any Transformer-based model inherently has a bottleneck, that is, a series of tokens converted from input images are given as input. Transformer's self-attention mechanism iteratively learns feature representation by associating a token with all other tokens, which also leads to a quadratic relationship between the model's complexity and the input tokens' number. This quadratic complexity prevents the vision Transformer-based backbone network from modeling high-resolution images, and the high computational cost poses a challenge for applying it to edge devices.

This study proposed two methods of linear attention mechanism and token pruning for improving Transformer models. Most of the existing Transformer models can be directly introduced to minimize the computational cost and speed up the model inside and outside respectively. Finally, the combination of the two methods obtained an e-attention. Overall, after changing the attention mechanism to linear alone, the FLOPs index is improved by about 50%–60%. After introducing the pruning module alone, the FLOPs index is increased by about 30%–40%. After introducing both improvements at the same time, the improvement ranges from 60% to 70%.

The performance drop of changing to linear attention mechanism alone is between 1.5% and 2.5%. After introducing the model lightweight token pruning module alone, the performance of the model drops between 0.2% and 0.5%, which is far less than the performance drop caused by the improvement of the attention mechanism. After introducing the two improvements, the overall model performance declines in the range of about 1.5%–3%.

For the target detection task, the change rule of the index mAP is the same as the index ACC in the image classification experiment, and it has a certain degree of decline. Because the improvement itself is to accelerate the model from the inside and outside, and this improvement is at the expense of the performance of the model. The experimental data showed that after the introduction of token pruning alone, the performance decline ranges from 0.5% to 1%. After introducing linear attention alone, the performance decline ranges from 1.5% to 2%. After the introduction of high-efficiency attention, the range of performance degradation is about 3%, which is within the acceptable range. Experiments show that after e-attention is introduced into the Transformer model, the calculation amount can be reduced by 60%–70% in image classification or target detection tasks, and the performance can be reduced by about 1.5%–3%.

This article accelerates the Transformer model from the perspective of linear attention and token pruning. For the lightweight pruning method of the Transformer model, in addition to considering the token dimension, pruning can also be considered from the attention head, neuron and other dimensions. Therefore, the pruning method can be combined with other dimensions to further accelerate the Transformer model in subsequent work. In addition, there are currently some works to accelerate Transformer from the perspective of distillation and quantization, which can also be tried for further research.

### Funding

This work was supported by the Sichuan Science and Technology Program (2021YFQ0003, 2023YFSY0026, 2023YFH0004). The funders had no role in study design, data collection and analysis, decision to publish, or preparation of the manuscript.

### Grant Disclosures

The following grant information was disclosed by the authors:
Sichuan Science and Technology Program: 2021YFQ0003, 2023YFSY0026, 2023YFH0004.

### Competing Interests

The authors declare there are no competing interests.

### Author Contributions

- Wenfeng Zheng conceived and designed the experiments, performed the experiments, performed the computation work, prepared figures and/or tables, authored or reviewed drafts of the article, and approved the final draft.
- Siyu Lu performed the experiments, prepared figures and/or tables, authored or reviewed drafts of the article, and approved the final draft.
- Youshuai Yang analyzed the data, performed the computation work, authored or reviewed drafts of the article, and approved the final draft.
- Zhengtong Yin analyzed the data, authored or reviewed drafts of the article, and approved the final draft.
- Lirong Yin analyzed the data, authored or reviewed drafts of the article, and approved the final draft.

### Data Availability

The code is available at Zenodo: Zheng, W., Lu, S., Yang, Y., Yin, Z., & Yin, L. (2023). Lightweight Transformer Image Feature Extraction Network. Zenodo. https://doi.org/10.5281/zenodo.10039236.

The raw data, ImageNet (ILSVRC2012), is available at https://www.image-net.org/challenges/LSVRC/2012/index.php#

The Common Objects in Context (COCO) dataset is available at https://cocodataset.org/#download.

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
