# Peer review of "Lightweight transformer image feature extraction network"

_PeerJ Computer Science, doi:10.7717/peerj-cs.1755_

## Round 0.1 · original submission · Minor Revisions

The reviewers have suggested minor changes to the paper.

So, the authors are advised to address the comments and carefully prepare a revised version while addressing the comments.

Reviewer 1 ·

Basic reporting

While the overall quality of the work appears great, there are distinguishable gaps in the coherence and flow of the content in certain sections. The authors have maintained a good level of explanation throughout but there are some improvements that can make the paper better. Specifically, the authors presume a high degree of familiarity with baseline models with the readers. While this is partly true, a little explanation of the models, mainly focusing on the components that authors compare throughout the paper, would bring more clarity.

Experimental design

The experimental design is sound and simple. The explanation is simple but there is some redundant information. For example, there is no need to introduce equation (15) again in the manuscript.

The paper does not cite the original attention framework from Vaswani et al. "Ashish Vaswani, Noam Shazeer, Niki Parmar, Jakob Uszkoreit, Llion Jones, Aidan N Gomez, Łukasz Kaiser, and Illia Polosukhin. Attention is all you need. In Advances in neural information processing systems, pages 5998–6008, 2017."

Validity of the findings

Combining the tables that compare the proposed approach with other methods into a single table alongside the comparisons of existing methods would enhance the readability of the paper, as currently, they are separated into different tables, which can be confusing. The experiments hold true for the dataset the authors have used i.e. ImageNet 1000 categories dataset. The paper title "Lightweight transformer image feature extraction network" would not hold true if the comparison is just made on the ImageNet dataset. In recent literature where similar research is done, it is standard practice to compare with more datasets. For example, the paper has primarily made comparisons with DeiT. The paper introducing the DeiT framework has a comparison with seven image datasets.

Additional comments

There is some typo in one of the references.

Touvron, H.; Cord, M.; Douze, M.; Massa, F.; Sablayrolles, A.; Jegou, H., Training data-efficient image transformers & distillation through attention. PMLR 2021.

It should be
Touvron, H.; Cord, M.; Douze, M.; Massa, F.; Sablayrolles, A.; Jegou, H., Training data-efficient image transformers & distillation through attention, In International Conference on Learning Representations (ICLR), PMLR 2021.

Cite this review as

·

Basic reporting

The article is as per the standards of Peerj journal

All the experiment results, graphs, tables, results, and research conducted are novel.

The paper needs to be converted into PeerJ manuscript format.

The paper needs to have related work/research

it would have been better if the Discussion and results were separated in the manuscript.

The attempts made on image feature extraction using AI techniques are outstanding.

Author jumped from Section 2 to Section 4

Experimental design

the research design is good and flow charts are completely aligned with the research conducted

Validity of the findings

The research conducted and findings are novel and outstanding

Additional comments

The article is as per the standards of Peerj journal

All the experiment results, graphs, tables, results, and research conducted are novel.

The paper needs to be converted into PeerJ manuscript format.

The paper needs to have related work/research

it would have been better if the Discussion and results were separated in the manuscript.

The attempts made on image feature extraction using AI techniques are outstanding.

Author jumped from Section 2 to Section 4

Cite this review as

Reviewer 3 ·

Basic reporting

1.This article is written in English and clear, unambiguous, technically correct text.

2.The article is having sufficient introduction and literature.
3. .The article is having figures and results.

4. There is no algorithm to the proposed technique. So, algorithm is required to the proposed technique.
Include it.

Experimental design

1. This article contains research within Aims and Scope of the journal.
2. The submission of article clearly defined the research question, which is relevant and meaningful.

Validity of the findings

New findings are available in this article. Which is useful to the society.

Additional comments

1.There is no algorithm to the proposed technique. So, algorithm is required to the proposed technique.
Include it.

2. In results section, there is no table which contains all results with comparison of other existing techniques. Include it.

3. In the conclusion part, time complexity of the proposed technique and other existing techniques are not mentioned.

Cite this review as

---

## Round 0.2 · accepted · Accept

After consideration of the comments from all the reviewers, I am convinced that the authors have addressed the concerns of the reviewers in the revised version.

Although one of the three independent reviewers feel that his/her concerns were not addressed, upon inspection, I can see that these comments are either addressed or the authors have provided the required information in a different style. For example, instead of providing a stand-alone algorithm, the authors have explained the methodology and supported the method with equations. Similarly, the tables do provide useful comparison, though the presentation style might not be as per the requirements of reviewer 3.

So, in summary, following the recommendations from the reviewers, I am accepting this paper.

Reviewer 1 ·

Basic reporting

The revision is great and addresses my concerns. It should be now suitable for publication.

Experimental design

No comments. All revision requests are made or addressed.

Validity of the findings

No comment.

Cite this review as

·

Basic reporting

1)This paper contains unique contents and research conducted is novel.
2) The submitted article is within our journal’s scope
3)The title idiom is concise and clear with the expected work
4)The author presented the main objective of the work properly in abstract section with suitable keywords
5)The author narrated the introduction section neatly to the planned work

Experimental design

1) A specific, clear, and predictive statement about the possible outcome of a scientific research study reported by the author
2) The author discussed about obtained results and associated details
3) Overall, Scientometric values are well presented in their designed work and Usefulness of this paper is worthy to publish
4) Conclusion and Limitations of the work derived and constructed well by the author and Authors’ contribution to the field of knowledge is appreciable
5) References are adequate, and mentioned appropriately

Validity of the findings

1)Research objective and findings having adequate particulars
2) The methods and techniques employed appropriately by the author
3) The findings and research conducted is novel

Additional comments

The paper can be accpeted with no further revision
Author addressed all the comments from the reviewer
paper need to be converted in Journal format
Abstract begin with "I" need to remove that
Reference [29 30] should be [29-30]

Cite this review as

Reviewer 3 ·

Basic reporting

I am not happy with author responses. The author didn't addressed properly to any of my questions.

Experimental design

NA

Validity of the findings

All underlying data have been provided.

Additional comments

I am not happy with author responses. The author didn't addressed properly to any of my questions.

My suggestions are very important to the manuscript.

Author needs to revise the manuscript as per my comments/suggestions.

Cite this review as